# Rethinking Local Branching: A Reinforcement Learning Approach to Neighborhood Control

## Abstract

For Mixed-Integer Linear Programming (MILP), the Local Branching (LB) heuristic is a well-established local search technique. However, its performance is highly sensitive to the neighborhood size—a parameter known to be instance-dependent. While recent learning-based methods aim to predict this numerical parameter, they often require extensive offline training data. This work introduces a novel approach that reframes neighborhood control in LB. Instead of predicting a size parameter, we learn a policy to select a subset of variables to which the LB constraint is applied. Our framework operates in two stages: first, we model the MILP instance as a graph and apply community detection to partition variables into structurally meaningful clusters, which serve as candidate neighborhoods. Second, a reinforcement learning (RL) agent dynamically selects the number of clusters to explore per iteration. Variables within chosen clusters are subjected to the LB constraint, while others are temporarily fixed. This results in an adaptive LB scheme where neighborhoods are defined by structural properties and dynamically scoped via RL—rather than by a single numerical parameter. Computational experiments demonstrate that our method automates neighborhood design without prior data collection. Evaluations across diverse MIP problems show that the proposed framework consistently outperforms state-of-the-art learning-based LB models and the open-source solver SCIP.

## 1 Introduction

Combinatorial optimization problems are ubiquitous in domains such as supply chain management, production scheduling, and network design. Mixed-Integer Linear Programming (MILP) constitutes a foundational modeling framework for addressing such problems. Despite substantial performance improvements modern MILP solvers like SCIP and CPLEX, the NP-hard nature of these problems presents formidable computational hurdles, particularly for large-scale and structurally complex instances. In practical applications, obtaining high-quality feasible solutions within a limited time budget is often paramount (Helber & Sahling, 2010; Chen, 2015; Gansterer et al., 2021; Qin et al., 2024). Consequently, the development of efficient heuristics to accelerate the discovery of superior solutions remains a central research direction in operations research. While the spectrum of heuristic algorithms for MILP is broad, this work focuses on Local Branching (LB), a prominent methodology within the class of Neighborhood Search (NS) algorithms. LB is characterized by its strategic utilization of MILP solvers as "black-box" tools to explore well-defined, mathematically-constrained neighborhoods.

The Local Branching (LB) heuristic, introduced by Fischetti & Lodi (2003), operates by adding a local branching constraint to the model, which effectively restricts the search neighborhood defined by the Hamming distance around the current solution. However, the performance of LB is highly sensitive to the setting of its key parameter, the neighborhood radius $k$. The selection of $k$ presents a significant dilemma: a value that is too small may trap the search in a local optimum, while an overly large $k$ renders the subproblem computationally expensive, thereby undermining the heuristic's purpose of rapid iteration. Recognizing this, a substantial body of research has focused on finding an appropriate value for $k$. The original LB algorithm (Fischetti & Lodi, 2003) initializes $k$ with a small, conservative value. While this approach yields a series of easy-to-solve subproblems, it often

leads to only marginal progress in the objective, leaving significant room for improvement. Fischetti & Monaci (2014) observe that ad-hoc tuning of the neighborhood size can markedly improve the performance of Local Branching. More recently, the increasing availability of real-world datasets has motivated research into data-driven machine learning approaches for accelerating MIP solving (Nair et al., 2020; Ding et al., 2020; Etheve et al., 2020; Qu et al., 2022; Scavuzzo et al., 2022; Zhang et al., 2023; Parsonson et al., 2023; Liu et al., 2024; Zhang et al., 2024). Liu et al. (2022) devised a learning-based framework to guide the LB search. Their two-phase approach first uses a scaled regression model to predict an initial $k$ and then employs a reinforcement learning strategy to dynamically adapt it. However, their experiments reveal that the overall performance is heavily reliant on the quality of the initial $k$ predicted by the regression model. This still necessitates a costly and onerous process of offline data collection, and the manual process of identifying a suitable sequence of $k$ values for diverse problem instances remains a time-consuming bottleneck. In light of these challenges, our aim is to design a self-learning algorithm that does not require a priori dataset collection and construction, thereby offering a more automated and general solution.

To address the aforementioned challenges, we introduce a novel approach that fundamentally reframes the control of the Local Branching heuristic. We shift the focus from the difficult problem of parameter learning to the more structured problem of variable set selection. Our core idea is to make the search neighborhood aware of the problem's intrinsic structure. To this end, we leverage a graph-based representation of the MILP instance, upon which we apply community detection algorithms to automatically partition variables into structurally-related clusters. These clusters become the fundamental building blocks for our search. We then employ a Reinforcement Learning (RL) agent that learns a sophisticated policy to dynamically select a combination of these clusters for exploration at each iteration. The LB constraint is applied exclusively to the variables within the chosen clusters, while the remaining "non-critical" variables are temporarily fixed. This transforms the neighborhood radius $k$ from a predefined parameter into a dynamic consequence of the agent's policy, determined by the size of the selected variable set. The result is an automated, self-learning framework that intelligently designs its search strategy based on problem structure, eliminating the need for costly offline data collection and manual tuning.

## 2 RELATED WORK

Local Branching is a seminal Math-Heuristic that leverages a general-purpose MIP solver to efficiently explore mathematically-defined neighborhoods (Fischetti & Lodi, 2003). The empirical effectiveness of LB has been demonstrated on many NP-hard problems. On benchmarks such as the capacitated fixed-charge network design problem, its performance has even surpassed that of many domain-specific heuristics (Rodríguez-Martín & Salazar-González, 2010). Since its inception, the theory and application of LB have been significantly extended. Its role has expanded from an improvement heuristic, which requires an initial feasible solution, to a feasibility heuristic capable of starting from an infeasible point, by integrating with techniques like the Feasibility Pump (Fischetti & Lodi, 2008). Concurrently, its applicability has been extended from its original focus on binary variables to broader models involving general integer variables (Yaghini et al., 2013), and it has been successfully applied to non-convex Mixed-Integer Non-Linear Programming (MINLP) problems (Nannicini et al., 2008). Methodologically, the flexibility of LB allows it to be embedded as a core component within other frameworks, such as Benders Decomposition (Rei et al., 2009) and Variable Neighborhood Search (VNS) (Hansen et al., 2006). More recently, Liu et al. (2022) proposed an adaptive strategy that leverages machine learning to directly learn and adjust the neighborhood size parameter $k$. In general, the selection of an appropriate $k$ value significantly impacts solving efficiency, but determining the optimal $k$ typically requires extensive problem-specific manual tuning. In this paper, in instead of directly optimizing $k$, we focus on identifying branching variable groups based on problem structure, thereby indirectly determining the effective searching neighborhood radius $k$. This fundamental reframing from a scalar parameter control problem to a combinatorial control problem enables our method to exploit problem structure more effectively and operate without reliance on pre-trained, data-intensive models.

Our method of selecting a subset of variables for partial branching and fixing is conceptually rooted in the "destroy-and-repair" framework of Large Neighborhood Search (LNS). As a metaheuristic, LNS iteratively improves a solution by first destroying a portion of it and then repairing that part. A central challenge in LNS is the design of an effective destroy operator, i.e., a policy for choos-

ing which variables to relax, particularly in large-scale problems. Recently, RL has gained traction for learning adaptive destroy operators. For instance, Song et al. (2020) developed an RL-based approach to learn a neighborhood selection policy, showing considerable improvements in solution speed. Likewise, Wu et al. (2021) integrated RL with LNS to dynamically adjust the neighborhood size, moving beyond fixed-size strategies. These advances highlight RL's ability to dynamically steer the search process in LNS, yielding gains in efficiency and solution quality. In contrast to existing learning-based neighborhood control methods, our framework integrates the generation and selection of neighborhoods within an adaptive self-learning framework. We first introduce a problem-agnostic strategy that uses graph community detection to form structurally meaningful variable clusters as candidate neighborhoods. We then train an RL agent to learn a policy that dynamically selects and combines these structurally-informed clusters at each iteration.

## 3 PRELIMINARIES

**Mixed-Integer Linear Programming (MILP)**   In this paper, we consider a generic MILP problem with 0-1 variables of the form:

$$
\begin{aligned}
(P) \quad \min \quad & c^T x \\
\text{s.t.} \quad & Ax \geq b, \\
& x_j \in \{0,1\}, \quad \forall j \in \mathcal{B} \neq \emptyset, \\
& x_j \geq 0, \qquad \forall j \in \mathcal{C}.
\end{aligned}
\tag{1}
$$

where $A$ is a $m \times n$ input matrix, and $b$ and $c$ are input vectors of dimension $m$ and $n$, respectively. Here, the variable index set $\mathcal{N} := \{1, \ldots, n\}$ is partitioned into $(\mathcal{B}, \mathcal{C})$, where $\mathcal{B} \neq \emptyset$ is the index set of the 0-1 variables, while the possibly empty sets $\mathcal{C}$ index the continuous variables.

**Local Branching**   The Local Branching approach aims to find an improved solution $x$ that is in the vicinity of a given feasible solution $\bar{x}$. To achieve this, it imposes an additional linear inequality, known as the local branching constraint, on the original problem $(P)$. This constraint defines the neighborhood of the search:

$$
\Delta(x, \bar{x}) := \sum_{j \in \bar{S}} (1 - x_j) + \sum_{j \in \mathcal{B} \setminus \bar{S}} x_j \leq k,
$$

where $k$ is a user-defined integer that sets the maximum allowable deviation from $\bar{x}$. The term $\Delta(x, \bar{x})$ represents the total number of binary variables that change their value. Denoted $\bar{S} := \{j \in \mathcal{B} : \bar{x}_j = 1\}$ as the binary support of $\bar{x}$, this can be calculated by summing the variables that switch from 1 to 0 (the first term) and those that switch from 0 to 1 (the second term). As its name implies, this inequality can be integrated as a branching rule in an enumerative scheme.

## 4 METHODOLOGY

Our methodology introduces a novel, two-stage framework to create an adaptive Local Branching heuristic. The first stage is a preprocessing step that leverages graph clustering to discover the intrinsic structure of a MILP instance, generating a set of high-quality candidate neighborhoods. The second stage employs an RL agent to dynamically control a search policy over these neighborhoods. The overall architecture is illustrated in Figure 1.

### 4.1 STRUCTURE-AWARE NEIGHBORHOOD GENERATION

Traditional LNS methods often rely on handcrafted, problem-specific rules or simplistic random partitioning to define variable subsets. Such strategies lack generalizability and frequently fail to exploit the complex coupling relationships inherent in the MILP's structure. To overcome this limitation, we perform a problem-agnostic variable clustering procedure.

The initial step is to represent the MILP instance as a bipartite graph, where one set of nodes corresponds to the decision variables and the other to the constraints and objective function. On this

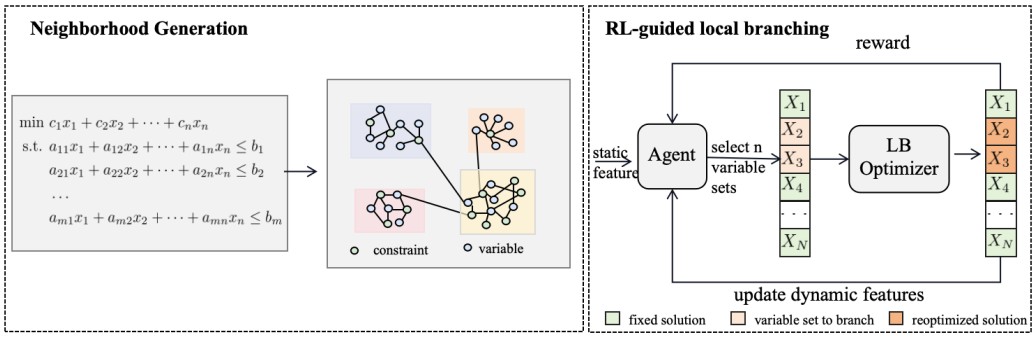

Figure 1: An overview of the proposed framework. Stage 1 (Neighborhood Generation): The MILP is converted to a graph and clustered to generate candidate neighborhoods. Stage 2 (RL-Guided Local Branching): An RL agent interacts with the solver, dynamically selecting neighborhoods to guide the search.

graph, we apply the Louvain community detection algorithm (Blondel et al., 2008). The Louvain method is an efficient algorithm that iteratively optimizes a "modularity" metric to discover tightly-knit communities of nodes. On our bipartite graph structure, the algorithm automatically classifies variables by iteratively evaluating the modularity gain from moving each node into the community of its neighbors. Since modularity rewards dense intra-community connections, variable nodes that frequently co-occur in the same constraints are naturally drawn together. Their shared constraint nodes act as bridges, making it mathematically advantageous from a modularity perspective to group them into the same community. For instance, if variables $x_1$, $x_2$ both appear in the same constraints, the algorithm will identify a significant modularity gain by placing $x_1$, $x_2$ and these shared constraint nodes into a single, cohesive cluster. This entire process is driven by the graph's topology, and crucially, it does not require a predefined number of clusters, allowing it to automatically identify the groups of variables that are naturally coupled within the problem's structure. The output is a collection of clusters, where each cluster defines a candidate neighborhood of variables that exhibit structural cohesion. This process provides a set of meaningful, structurally-aware neighborhoods that serve as the foundational building blocks for the subsequent search stage.

## 4.2 RL-Guided Local Branching

With a set of candidate neighborhoods established, the next step of our approach is to train a Reinforcement Learning agent that learns a policy to dynamically control the search. We formulate this sequential decision problem as a discrete-time Markov Decision Process (MDP).

**MDP Formulation.** An episode consists of the iterative process of solving a single MILP instance. The components are defined as follows:

- **State** ($s_t$): A state capture both the static features of the instance and the dynamic features of the solving process. Following Gasse et al. (2019), we represent the state $s_t$ as a bipartite graph embedding that includes static features of the variables and constraints, as well as dynamic features from the current solving state (including incumbent solution $x_t$, best solution $x^*$ et al.).

- **Action** ($a_t$): To enhance the ability of generalization, the agent learns a *relative action space*. The action $a_t$ is a discrete choice from the set $A = \{-\Delta, 0, +\Delta\}$, where $\Delta$ is a step size. This action adjusts the number of neighborhoods to be explored in the next iteration, $n_{t+1} = n_t + a_t$, rather than deciding an absolute value.

- **Reward** ($r_t$): The reward function is designed to balance solution quality and computational cost. It is defined as the normalized rate of improvement: $r_t = \frac{\Delta_{\text{obj}}}{\Delta_t}$, where $\Delta_{\text{obj}} = \frac{c^T x_t - c^T x_{t+1}}{|c^T x_0|}$ is the normalized objective improvement relative to the initial solution $x_0$, and $\Delta_t$ is the elapsed time.

**Neighborhood Selection and local branching.** At each step $t$, the RL agent takes action $a_t$ to determine the number of neighborhoods, $n_t$, to explore. To select the specific $n_t$ neighborhoods, we employ a history-aware exploration strategy to promote diversification. The probability of selecting a neighborhood $i$ is inversely weighted by its historical selection count: $P(i) \propto \frac{1}{\text{count}_i + 1}$. This sampling method ensures a balanced exploration of all candidate neighborhoods, preventing premature convergence. Once the variable clusters for branching are chosen, the local branching constraint is applied. Instead using a fixed neighborhood radius $k$, we set it dynamically based on the current incumbent solution $\bar{x}$ within the selected variable set $\mathcal{B}$, where $\bar{\mathcal{S}} = \{j \in \mathcal{B} \mid \bar{x}_j = 1\}$. The rule is defined as: $\Delta(x, \bar{x}) := \sum_{j \in \bar{\mathcal{S}}} (1 - x_j) + \sum_{j \in \mathcal{B} \setminus \bar{\mathcal{S}}} x_j \leq k$, where $k \propto \left( \sqrt{|\bar{\mathcal{S}}|} + \sqrt{|\mathcal{B} \setminus \bar{\mathcal{S}}|} \right)$

This heuristic design is critical for maintaining a balance between exploration scope and computational tractability. It scales sub-linearly with the problem size, which prevents the subproblem from becoming computationally prohibitive when the branching set is large, while simultaneously providing a sufficiently permissive search radius to escape local optimal when the set is small. In addition to the adaptive radius, the computational budget for each LB subproblem is also dynamically adjusted. The time limit for each iteration is determined by the number of selected variable clusters, $n$, according to the rule $T_t = T_0 \cdot (n_t/n_0)^\alpha$, where $T_0$ is the initial time limit, $n_0$ is the initial number of branching neighborhoods, and $\alpha$ is a scaling exponent. The intuition is that the difficulty of a subproblem is correlated with the number of variables being branched upon. By allocating more computational time to these larger, potentially more difficult subproblems and enforces rapid iterations for smaller ones, further optimizing the overall search efficiency.

**Policy Parametrization and Updating.** To learn the policy $\pi_\theta(a_t|s_t)$, we parametrize it using the Graph Convolutional Network (GCN) (Kipf, 2016), which is a GNN variant and ideally suited for this task due to its ability to process graph-structured inputs of arbitrary size. The GCN-based policy network takes the current state graph as input and outputs a probability distribution over the actions, from which $a_t$ is sampled. The details of policy training are given in Algorithm 1. For a given instance, the algorithm operates iteratively to solve the problem while gathering training samples. At each step within the solving loop, it first constructs a state graph $s$ that encodes both static features of the instance $S^p$ and dynamic features $S^d$ from the current search process (Line 4). This state is then fed to the policy $\pi_\theta$ to select an action $n$, which determines the number of neighborhoods to explore (Line 5). Subsequently, the problem is re-optimized by performing local branching on the chosen variable subsets (Lines 7-8). This interaction generates an experience tuple $(s, n, r)$ (Line 12). After the termination conditions for the instance are met, we employ the REINFORCE policy gradient method (Sutton et al., 1999) to update the policy network's parameters $\theta$, leveraging the complete trajectory of these experience tuples.

# 5 EXPERIMENTS

## 5.1 EXPERIMENTAL PROTOCOLS

In this section, we compare our approach against several baselines on a suite of MILP benchmarks, using SCIP as the underlying MILP solver.

**Instance generation.** We evaluate our method on three well-known classes of MILP benchmarks: Set Covering (SC), Maximum Independent Set (MIS), and Combinatorial Auction (CA). To ensure a fair and reproducible comparison, instance generation follows the procedures outlined in Liu et al. (2022) and Gasse et al. (2019). Specifically, we generate SC instances with 2000 columns and 5000 rows, MIS instances on Barabási–Albert random graphs with 1000 nodes, and CA instances with 4000 items and 2000 bids. The details of instances are shown in Table 1. For each problem class, we generate 160 instances for training, 40 for validation, and 40 for testing. To assess the generalization capability of our learned policies, we also generate an additional set of 40 larger instances for each problem class, where the number of variables and constraints are doubled. To further validate the practical efficacy of our approach, we test its performance on the MIPLIB dataset (Gleixner et al., 2021). Notably, instead of training our model on this datasetwe directly apply the policies trained on the other three benchmark classes to the MIPLIB instances. This setup serves as a rigorous assessment of our model's generalization capabilities.

---

**Algorithm 1** RL-Guided Branching Algorithm

---

**Input:**
    Instances set $P$; initial branching number $n_0$; total time limit $T$; total iterations limit $Iter$; initial branching policy $\pi_\theta$;
**Output:**
    branching policy $\pi_\theta$
1: **for** each instance $p \in P$ **do**
2:     $t \leftarrow 0$, $i \leftarrow 0$, $n \leftarrow n_0$
3:     Initialize the dynamic features $S^d$
4:     Obtain clustered variable set $X = \{X_i\}$
5:     **while** $t \leq T$ **and** $i \leq Iter$ **do**
6:         Construct state representation $s \leftarrow \text{Graph}(S^p, S^d)$
7:         Sample branching number $n \leftarrow \pi_\theta(s)$
8:         Select branching set $\mathcal{B} \subseteq X$ with $|\mathcal{B}| = n$
9:         Apply local branching to $\mathcal{B}$, fixing variables in $X \setminus \mathcal{B}$
10:        Solve subproblem to obtain new incumbent $(\bar{x}, \overline{obj})$
11:        **if** $\overline{obj}$ improves $obj^*$ **then**
12:            Update incumbent: $(x^*, obj^*) \leftarrow (\bar{x}, \overline{obj})$
13:        **end if**
14:        Compute the reward $r$ and collect tuple $(s, n, r)$
15:        Update elapsed time $t$, update $i \leftarrow i + 1$
16:        Update dynamic state $S^d$
17:    **end while**
18:    Update policy parameters $\theta$ for the policy $\pi_\theta$
19: **end for**
20: **return** $\pi_\theta$

---

Table 1: The protocol of the binary integer programming problem.

| Size | Set Covering | | Independent Set | | Combinatorial Auction | |
|---|---|---|---|---|---|---|
| | columns | rows | columns | average rows | columns | average rows |
| Small | 2000 | 5000 | 1000 | 4000 | 2000 | 3500 |
| Large | 4000 | 10000 | 2000 | 8000 | 4000 | 7000 |

**Features.** The state representation for our RL agent combines both static and dynamic features of the MIP instance. Static Features: To represent the problem's intrinsic structure, we extract features from the variables and constraints to construct a bipartite graph representation, similar to Gasse et al. (2019). We also include the objective coefficients and the solution values from the initial LP relaxation as part of the static feature set. Dynamic Features: To capture the progress of the search, the dynamic features include the variable values in the current incumbent solution, the current best solution, the cumulative elapsed time and so on. These static and dynamic features are concatenated to form the feature vectors for the nodes in the bipartite graph. A detailed description of all features shows in Table 2.

**Hyperparameters.** We use SCIP (v8.1.0) Achterberg (2009) as the underlying MILP solver, which also serves as a primary baseline. All experiments are conducted on a machine with an Apple M3 8-core CPU and 16GB of RAM. For each benchmark class, our RL agent is trained for 20 episodes. The total time limit for solving each instance is set to 300s.We employ an early stopping criterion that terminates the procedure if the objective value shows no improvement beyond a set threshold for 20 consecutive iterations. Additionally, the time limit for each local branching step is dynamically adjusted based on the size of the branching variable set, with detailed configuration provided in the Appendix.

**Evaluation.** As the NP-hard instances are too large to solve in a reasonable time, we evaluate methods in there metrics: average PrimalBound, average PrimalGap and average PrimalIntegral

Table 2: Feature description

| Name | Type | Feature | Description |
|------|------|---------|-------------|
| Variable Features (V) | static | Objective | Objective coefficient |
| | | Is_type_binary | Binary variable indicator |
| | | Is_type_continuous | Continuous variable indicator |
| | | Has_lower_bound | Lower bound existence indicator |
| | | Has_upper_bound | Upper bound existence indicator |
| | | Lower_bound | Lower bound value |
| | | Upper_bound | Upper bound value |
| | | Cluster_category | Category identifier |
| | dynamic | Incumbent_solution | Current incumbent solution value |
| | | Best_solution | Best known solution value |
| | | Is_branching | Branching candidate indicator |
| | | Branching_times | Number of times variable branched on |
| Constraint Features (C) | static | LHS | Left-hand side value |
| | | RHS | Right-hand side value |
| Edge Features (E) | static | Coef | Constraint coefficient |

within the solving time limit. The PrimalBound is the objective value $o^*$ of the best feasible solution found within the limited time. The PrimalGap reflects the difference between the solution of a method to the best one found by all methods. For a given instance, the primal gap is defined as: $PrimalGap(t) = \frac{|o^* - p^*|}{\max(|o^*|, |p^*|)}$, where $p^*$ is the objective value of the best-known feasible solution for that instance. Also, to capture both the quality of the bounds and the speed at which they are found, we compare the cumulative progress within a fixed time horizon $T$. The primal integral metric is defined as $PrimalIntegral = \int_0^T o(t)^* dt$, where a lower value indicates better overall performance within the time limit $T$. This metric effectively rewards algorithms that make rapid and substantial progress on the primal bound early in the search process.

## 5.2 COMPARATIVE EXPERIMENT

**Baselines.** We compare our method with four algorithms:

- **SCIP (v8.1.0):** State-of-the-art open-source solver with default settings.

- **LB-SRMRL:** A LB version in Liu et al. (2022), which trains a regression model to predict an optimal initial radius $k$, and subsequently uses reinforcement learning to dynamically adjust it.

- **SARLB-RL-off:** Our method SARLB replaces the RL agent with a random action selector.

- **SARLB-SA-off:** Our method SARLB disables the neighborhood generation, which means grouping variables randomly.

**Comparative analysis.** Table 3 depicts the overall performance on the three standard benchmark problems. Among the learning-based methods, our SARLB framework consistently outperforms the LB-SRMRL baseline across all three problem classes. The final PrimalGaps achieved by SARLB are substantially smaller, indicating that our method consistently yields higher-quality solutions within the given time limit. Furthermore, the PrimalIntegral metric reveals a more pronounced advantage, SARLB consistently yields lower integral values, which signifies not only a superior final solution but also a significantly faster convergence towards it. This rapid convergence is visually corroborated by the plots in Figure 2. Across all three benchmarks, the SARLB curve (solid red line) demonstrates a significantly steeper initial descent and converges to a lower terminal gap compared to the slower, more staggered progress of the LB-SRMRL baseline (dash-dotted purple line). These results empirically suggest that learning a policy to select structurally-aware variable subsets constitutes a more effective and efficient paradigm for neighborhood control than directly learning to tune a single numerical parameter, $k$.

Table 3: Evaluation results of different algorthoms on datasets.

| LB Methods | Set Covering | | | Independent Set | | | Combinatorial Auction | | |
|---|---|---|---|---|---|---|---|---|---|
| | PrimalBound | PrimalGap | PrimalIntegral | PrimalBound | PrimalGap | PrimalIntegral | PrimalBound | PrimalGap | PrimalIntegral |
| LB-SRMRL | 3,455.45 | 4.05% | 1,272,503.68 | -454.88 | 0.21% | -45,286.12 | -130,797.46 | 0.20% | -39,107,578.10 |
| SNLB | 3,354.24 | 1.03% | 852,284.12 | -455.80 | 0 | -45,508.13 | -130,962.61 | 0.06% | -39,727,655.49 |
| SARLB-RL-off | 3,418.48 | 2.54% | 1,026,686.66 | -453.75 | 0.45% | -44,995.57 | -130,403.53 | 0.59% | -38,993,685.21 |
| SARLB-SA-off | 3,405.41 | 2.28% | 1,013,318.27 | -455.77 | 0.01% | -45,208.60 | -130,933.49 | 0.08% | -39,075,489.19 |

**Ablation study.** To deconstruct the performance gains, we present an ablation study in Table 3 that verifies the effectiveness of its two key architectural components: the structure-aware neighborhood generation and the RL-guided control policy. The results unequivocally suggest that both components contribute positively to the overall performance, as disabling either one leads to a degradation in solution quality and convergence speed. Specifically, we observe that the performance drop is substantially more pronounced when the RL agent is disabled (SARLB-RL-off) compared to when the graph clustering is replaced with random partitioning (SARLB-SA-off). This may reveal that while the graph clustering provides a strong structural foundation, the ability of the online RL agent to drive deeper exploration from the iteratively updated policies is the more essential element of our framework's success. The convergence plots in Figure 2 provide a clear visual illustration of this disparity. Disabling the RL-guided control policy (SARLB-RL-off, dotted green line) results in a severe performance degradation across all benchmarks, unequivocally confirming that the dynamic learning agent is the primary driver of SARLB's success. Conversely, replacing the structure-aware neighborhood generation with random partitioning (SARLB-SA-off, dashed blue line) yields a more competitive performance, yet it is still consistently outperformed by the integrated SARLB model. This finding also validates the contribution of the graph clustering component; while the RL agent is paramount, furnishing it with structurally-meaningful candidate neighborhoods provides a distinct and valuable performance enhancement.

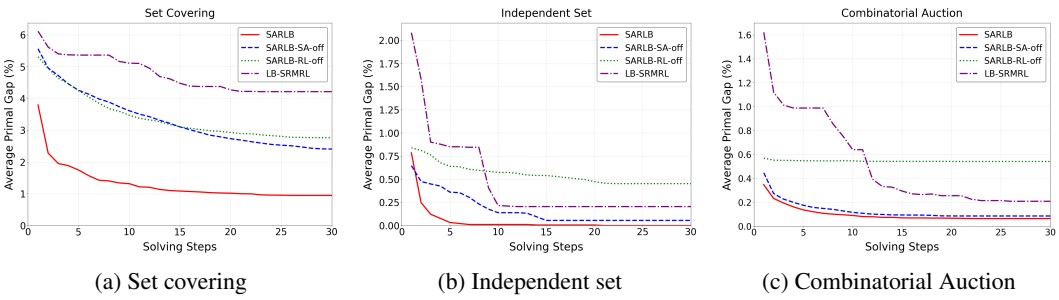

| (a) Set covering | (b) Independent set | (c) Combinatorial Auction |
|---|---|---|

Figure 2: The average primal gap curve during the solving process for three datasets.

**Generalization analysis.** To evaluate the generalization ability of our framework, we further test our method on larger instances and on the diverse MIPLIB 2017 benchmark. The overall results are gathered in Table 4. On the larger instance sets, our proposed method consistently outperforms the competing baselines in terms of both final solution quality and convergence speed, demonstrating superior generalization to increased problem scales. For the challenging MIPLIB dataset, a collection of real-world MILP instances, our method (using policies trained solely on synthetic data) also exhibits competitive generalization. The results show a notably lower PrimalGap compared to other learning-based methods like LB-SRMRL. Furthermore, the performance of SARLB with a 300s time limit is often competitive with, or even surpasses, that of SCIP running for 500s. This substantiates the efficacy of our instance-wise, higher-level control policy for neighborhood selection, emphasizing its versatility and robustness across diverse problem domains. A significant distinction from previous works is that our method learns a higher-level control policy at an instance-wise level for neighborhood selection, rather than making predictions on individual variables or a single numerical parameter, which contributes to its competitive generalization performance.

Table 4: Generalization ability on large datasets.

| LB Methods | Large Set Covering | | Large Independent Set | | Large Combinatorial Auction | | MIPLIB | |
|---|---|---|---|---|---|---|---|---|
| | PrimalBound | PrimalGap | PrimalBound | PrimalGap | PrimalBound | PrimalGap | PrimalBound | PrimalGap |
| LB-SRMRL | 2266.58 | 6.07% | -911.30 | 0.45% | -258,598.77 | 0.83% | 8,777,602,039 | 11.80% |
| SARLB | 2136.57 | 1.63% | -915.40 | 0 | -260,415.89 | 0.13% | 9,049,794,557 | 1.36% |
| SCIP(300s) | 2295.20 | 6.71% | -912.80 | 0.45% | -260,206.91 | 0.21% | 8,729,503,534 | 6.64% |
| SCIP(500s) | 2278.90 | 6.01% | -913.07 | 0.25% | -260,364.47 | 0.15% | 8,746,170,201 | 6.49% |

# 6 CONCLUSION AND OUTLOOK

In this work, we revisit the Local Branching paradigm through a machine learning lens, adding to a growing body of literature that uses ML/RL to speed up the solving process of MILPs. We reframe the classical challenge of tuning the neighborhood radius $k$ from a problem of parameter prediction or tunning into a higher-level problem of policy learning for variable set selection. Our framework consists of a two-phase strategy: a graph clustering stage to generate structurally-aware candidate neighborhoods, and a reinforcement learning phase to learn a dynamic policy for selecting these neighborhoods. Experiments on different MIP problems shows its effectiveness and superior generalization ability. The algorithm generalizes well not only to larger instances of the same class but also across diverse problem types from the MIPLIB benchmark, outperforming the solver and other learned-based algorithm.

For future research, a direction is to transcend the static nature of our neighborhood generation. In the current framework, variable clusters are determined a priori based on the initial problem structure. A more powerful method could involve dynamically updating these groupings by incorporating information that emerges during the search. This would yield neighborhoods that are not only structurally-aware but also search-state aware. Ultimately, by shifting the learning objective from size to composition, our work offers a robust and automated approach towards intelligent heuristic search for combinatorial optimization.

## ETHICS STATEMENT

This work adheres to the ICLR Code of Ethics and does not contain any studies with human participants performed by any of the authors. And we do not use any personally identifiable information, sensitive data, or datasets with inherent social biases.

## REPRODUCIBILITY STATEMENT

The experimental setup, including training steps, model configurations, and hardware details, is described in detail in the paper. The full code and dataset we used will be made publicly available on GitHub upon publication of this paper.

## LLM USAGE

During the preparation of this work, we used GPT-3.5 and Gemini-2.5-pro in order to improve the readability and language of this paper. After using this tool, we reviewed and edited the content as needed and take full responsibility for the content of the published article.

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

# A  APPENDIX

## A.1  TRAINING PROCEDURE

The reinforcement learning agent was trained for each of the three benchmark classes (Set Covering, Independent Set, and Combinatorial Auction) separately. For each class, the policy was trained for 20 episodes, where each episode consists of solving all 160 training instances once. The training was conducted on a single machine with an Apple M3 8-core CPU and 16GB of RAM. We used a grid-search procedure on the validation set to tune key hyperparameters for the GCN-based policy network, including the number of GCN layers ($n_l$), the initial learning rate (lr), and its decay schedule. The final hyperparameter choices are summarized in Table 5.

Table 5: The hyperparameters for evaluation.

| Hyperparameters | Set Covering | | Independent Set | | Combinatorial Auction | | MIPLIB |
|---|---|---|---|---|---|---|---|
| | Large | Small | Large | Small | Large | Small | |
| GCN Layers ($n_l$) | 7 | 5 | 7 | 5 | | 7 | 5 |
| Learning Rate (lr) | 0.01 | 0.001 | 0.01 | 0.001 | 0.01 | 0.001 | 0.01 |

Table 6: The hyperparameters for evaluation.

| Hyperparameters | Set Covering | | Independent Set | | Combinatorial Auction | | MIPLIB |
|---|---|---|---|---|---|---|---|
| | Large | Small | Large | Small | Large | Small | |
| Scaling Exponent $\alpha$ | 1.2 | 1 | 0.8 | 1 | 1.5 | 1 | 1 |
| Initial Time Limit $(n_0)T_0$ | 30s | 20s | 10s | 5s | 15s | 10s | 25s |
| Convergence Threshold $gap$ | 0.05% | 0.01% | 0.005% | 0.001% | 0.01% | 0.005% | 0.05% |

## A.2 EVALUATION PROCEDURE

During evaluation, we used the final trained policy for each benchmark class as described in Section
. The iterative search was terminated when either the total time limit of 300s was reached or if there
was no objective improvement of more than $gap$ for 20 consecutive iterations. For the dynamic
time limit rule, $T_t = T_0 \cdot (n_t/n_0)^{\alpha}$, we used a reference number of clusters $n_0 = |X|/2$. The
hyperparameter choices are summarized in Table 6.

## A.3 INPUT FEATURES

The state representation for our RL agent combines both static and dynamic features of the MILP
instance, as detailed in Section 5.1. The static features, which capture the problem's intrinsic struc-
ture, and the dynamic features, which capture the search progress, are used to construct the bipartite
graph representation. The code for computing these features is adapted from the open-source imple-
mentation of Gasse et al. (2019), which is publicly available at `https://github.com/ds4dm/`
`learn2branch`.

