# OpenReview forum: "Rethinking Local Branching: A Reinforcement Learning Approach to Neighborhood Control"
_ICLR.cc/2026/Conference — ICLR 2026 Conference Withdrawn Submission_

### Official Review · Reviewer_H5BD · 2025-10-29

**Soundness:** 2
**Presentation:** 3
**Contribution:** 1
**Rating:** 2
**Confidence:** 4

**Summary:**

This paper proposes a hybrid LB approach integrating two mechanisms, an overhead graph partition procedure, and an RL-based approach for determining the number of clusters of variables to be reoptimized. Experiments are conducted against SCIP and sota LB-based methods, demonstrating the effectiveness of their proposed method.

**Strengths:**

This work combines community partition techniques with reinforcement learning and improves the performance of LB.
The paper is well structured and presented with great clarity.

**Weaknesses:**

1. **Misleading story telling** The story begins with LB and discusses the importance of LB radius. However, the proposed approach is actually selecting a particular variable cluster to perform LB, **with the other variables fixed**. This is conceptually the large neighborhood search (LNS). More specifically, the proposed approach coincides with DINS~[1] by sharing the "hard fixing + soft fixing (LB)" scheme. Therefore, this work should be considered an improvement of LNS, rather than LB.
2. **Insufficient experimental setup** .
 3.  **Datasets**: When it comes to LNS, the goal is to tackle large-scale problems. The experiments consider problems with only thousands of variables. As a contradiction, CL-TLNS~[2] considers up to 100,000 variables.
4.  **Baselines**: As mentioned above, the proposed approach is LNS-based and should be compared against the other state-of-the-art LNS methods, such as CL-LNS~[3] and BTBS-LNS[4].

[1] Ghosh, Shubhashis. "DINS, a MIP improvement heuristic." International Conference on Integer Programming and Combinatorial Optimization. Berlin, Heidelberg: Springer Berlin Heidelberg, 2007.
[2] Liu, Wenbo, et al. "Mixed-Integer Linear Optimization via Learning-Based Two-Layer Large Neighborhood Search." THE 19TH LEARNING AND INTELLIGENT OPTIMIZATION CONFERENCE.
[3] Huang, Taoan, et al. "Searching large neighborhoods for integer linear programs with contrastive learning." International conference on machine learning. PMLR, 2023.
[4] Yuan, Hao, et al. "BTBS-LNS: Binarized-Tightening, Branch and Search on Learning LNS Policies for MIP." The Thirteenth International Conference on Learning Representations.

**Questions:**

1.  Is the runtime of the graph partitioning procedure included in the experimental results? Furthermore, when applied to significantly larger problems, could the overhead of partitioning outweigh the benefits it provides?
2.  How to determine the initial branching number $n_0$?
3.  The reward is defined as $\frac{\Delta_{obj}}{\Delta_t}$, while $\Delta_{obj}$ is clear, the interpretation of the solving time $\Delta_t$ is ambiguous. During training, the subproblem is likely solved exactly, so $\Delta_t$ corresponds to the time for exact solving. During inference, however, the subproblem solver should aim to quickly find high-quality solutions (e.g., using MIPFocus=1 in Gurobi), meaning $\Delta_t$ would reflect heuristic solution time. Thus, a large $\Delta_t$ during training may not correlate with a large $\Delta_t$ during inference, raising concerns about reward consistency across stages. Have the authors consider such issue or just consider exact solution during inference?

---

### Official Review · Reviewer_sUAf · 2025-10-30

**Soundness:** 2
**Presentation:** 2
**Contribution:** 2
**Rating:** 2
**Confidence:** 4

**Summary:**

This paper presents a novel reinforcement learning (RL) framework for adaptive neighborhood control in Local Branching (LB), a well-known heuristic for mixed-integer linear programming (MILP). The key idea is reframing neighborhood size tuning as a combinatorial policy learning problem. The empirical results demonstrate consistent improvements over baselines, including a learning-based LB variant and the SCIP solver.

**Strengths:**

None

**Weaknesses:**

1. The shift from predicting a scalar neighborhood size to learning a policy for selecting variable subsets is not new. "BTBS-LNS: Binarized-Tightening, Branch and Search on Learning LNS Policies for MIP ICLR 2025" uses similar techniques to create local branch constraints for wrongly-fixed variables. A more thorough background check and baseline comparison would be preferred to enhance the quality of this paper.
1. The paper lacks a theoretical discussion of why variable clusters from community detection are superior to random or handcrafted partitions. A brief analysis of the graph structure (e.g., modularity scores) could strengthen this claim.

**Questions:**

1. Could you compare the clustering method you used in this paper with random clustering? I would like to see if it truly helps.

---

### Official Review · Reviewer_Bgrp · 2025-10-30

**Soundness:** 1
**Presentation:** 2
**Contribution:** 2
**Rating:** 2
**Confidence:** 4

**Summary:**

This paper addresses the sensitivity of Local Branching (LB)—a classic heuristic for Mixed-Integer Linear Programming (MILP)—to its neighborhood size parameter, which is typically instance-dependent. Instead of predicting a scalar size, the authors propose a two-stage framework: first, they use community detection (e.g., Louvain) on a graph representation of the MILP to partition variables into structurally coherent clusters as candidate neighborhoods; second, a reinforcement learning (RL) agent dynamically selects how many clusters to activate per iteration, applying the LB constraint only to variables within those clusters. This yields an adaptive, structure-aware LB scheme that requires no offline training data. Experiments show consistent improvements over both a recent learning-based LB baseline and the open-source solver SCIP across diverse MIP benchmarks.

**Strengths:**

- The idea of structure-aware neighborhood generation—grouping tightly coupled variables via community detection—is insightful and aligns well with the combinatorial nature of MILP. The empirical gains support its practical value.
- The paper is well-organized and easy to follow, with a clear narrative flow from motivation to method to evaluation.
- The authors provide code, which significantly enhances reproducibility and transparency.

**Weaknesses:**

- The related work section is notably weak. Several recent and highly relevant learning-based large neighborhood search (LNS) and MILP solving methods are neither cited nor compared against, including:
  [1] Searching Large Neighborhoods for Integer Linear Programs with Contrastive Learning, ICML 2023.
  [2] ILP-FORMER: Solving Integer Linear Programming with Sequence to Multi-Label Learning, UAI 2024.
  [3] BTBS-LNS: Binarized-Tightening, Branch and Search on Learning LNS Policies for MIP, ICLR 2025.
  [4] Apollo-MILP: An Alternating Prediction-Correction Neural Solving Framework for Mixed-Integer Linear Programming, ICLR 2025.
  Given that Local Branching is a form of LNS, omitting comparisons with state-of-the-art LNS approaches undermines the empirical claims and contextual fairness.
- The methodological novelty appears limited. The structure-aware neighborhoods rely directly on the standard Louvain community detection algorithm without adaptation, and the RL-guided selection of the number of clusters closely resembles the framework in:
  [5] Learning to search in local branching, AAAI 2022.
  The key difference—selecting the count of neighborhoods rather than a size parameter—still requires a handcrafted rule to determine the actual cluster count, which diminishes the role of RL. As such, the contribution feels incremental rather than substantial.
- The experimental baselines are relatively narrow, comparing against only one learning-based LB method and SCIP. The absence of comparisons with recent SOTA LNS (e.g., [1]–[4]) makes it difficult to assess whether the observed gains stem from the proposed design or simply from weaknesses in the chosen baselines.

**Questions:**

- Were all MIPLIB instances used, or a filtered subset? If filtered, what was the selection criterion? Could the authors provide per-instance results to better illustrate performance variance?
- Why does the RL agent only decide the *number* of neighborhoods to activate, rather than directly selecting *which* clusters (or assigning selection probabilities)? This design seems to limit the agent’s influence—what motivated this choice?
- What motivated the focus on Local Branching, which has relatively few learning-based studies, rather than the broader and more active LNS literature?
- In the bipartite graph construction, both constraints and the objective function are placed on the same side. Are their feature representations aligned or normalized in a compatible way? Could this lead to representation imbalance?
- In Table 3, what does “SNLB” refer to? Please clarify the algorithm or provide a citation.
- In Figure 2, do all the compared methods start from the same initial solution? If not, the comparison may not be fully fair—could the authors confirm initialization consistency?

**I will actively engage in the rebuttal discussion. Should any of my concerns stem from misunderstandings, I welcome the authors’ clarification and will update my evaluation accordingly**.

---

### Official Review · Reviewer_WQ8V · 2025-11-01

**Soundness:** 1
**Presentation:** 2
**Contribution:** 1
**Rating:** 2
**Confidence:** 4

**Summary:**

This paper is about a metaheuristics, known as Local Branching (LB) that destroys/repairs part of the binary variables for mixed-integer programming.

LB is a well-known LNS meta-heuristics and ML+RL based methods have been extensively studied in this context.

The paper proposes "structure-aware" clustering of the standard bipartite variable/constraint graph to restrict the selection of variables to destroy within a cluster. Then an RL-based policy is trained to select the number of clusters to destroy. This effectively replaces the "k" parameter of the original Local Branching that decides the number of variables to destroy.

This method is evaluated on randomly generating instances and MIPLIB17.

**Strengths:**

The idea of higher-level decision making via clustering reduces the action space nicely.
Graph-based community detection algorithm seems quite practical
Method shows generalization in terms size and out of domain
Ablation on RL and SA is nice

**Weaknesses:**

My major concern with the paper is: it does not compare with the state-of-the-art/more recent works. Why did you compare with LB-Liu'22 and not more recent Local Branching Relaxation (CPAIOR'23).

LB-Relax is a much improved version of LB so it would be the right comparison to better evaluate the significance/value of the empirical results presented here.

Along those lines; there are also RL-variants that is not even mentioned here such as Searching Large Neighborhoods for Integer Linear Programs with Contrastive Learning (ICML'23).

Other immediately relevant but not cited works are:
- A general large neighborhood search framework for solving integer programs. (NeurIPS'20 Song et. al)
- Learning a large neighborhood search algorithm for mixed integer programs. (Sonnernat et. al.)
- Learning large neighborhood search policy for integer programming (NeurIPS'21, Wu et. al.)
- Balans (IJCAI'25)

In particular, Balans would make an excellent baseline comparison with LB-k1, LB-k2, etc.. are added as arms in an adaptive LNS running online driven by bandits to compare learning "clusters" (higher-level structure aware) against "k" (lower level variables)

- LB-relax is a must have comparison for this paper. It's unfortunately not considered.

- Why did you use the old MIPLIB17 and not the newer MIPLIB24?

- Why did you generate random instances and not use the Distributional MIPLIB that has the same problem domains (SCP, ISP, CA) and more like Vertex Cover, Generalized ISP, Load Balancing, Item Placement etc..

- The method clusters once at the beginning of the search, which remains completely static for the rest of the search. This seems like an opportunity to improve. Dynamic state-information can be valuable for adapting neighborhoods.

- The paper should make it more explicit that Local Branching only applies to Binary Variables. This is quite limiting for general Mixed-Integer Programming Problems.

- There are no runtime breakdowns or analysis of overhead of components; graph construction, neighborhood analysis, training, inference, mip solves per iteration etc.

- Overall this is an empirical paper with no theoretical analysis of convergence, sample complexity, or any guarantees. ML+RL combinations for primal heuristics (like Local branching) and MDP formulations have been studied extensively (although the paper omits)

- The current reward function seems simplistic (not necessarily a bad thing, just observation). Have you considered other factors like primal-dual gap, diversity of moves etc.

- The RL agent is trained on synthetic benchmarks and evaluated on similar distributions. How different is MIPLIB17 from those?

- I am surprised that there was no details on the Louvain algorithm or any other alternatives. The choice of Louvain for clustering the bipartite graph seems arbitrary. Why not the more stable Leiden cluster? Or more standard Hierarchical / Agglomerative Clustering. What resolution/gamma parameter is used for Louvain. And, how stable the Louvain clustering. node ordering affects the clusters, no? And that clustering setup stays "fixed" for the rest of the pipeline, so some sensitivity analysis is required here.

- I believe that the method has the potential to be solver-agnostic. The improvements relative to SCIP solver is nice, and it would strengthen the paper to show similar/better results with another solver, say Gurobi, so that we know the results obtained are invariant to the choice of SCIP. Btw, SCIP does employ a LNS inside the branch and bound tree. It is important to disclose whether this setting is used or not --which is currently not discussed in the paper.

**Questions:**

See above

---

### Note · Authors · 2025-11-16

I have read and agree with the venue's withdrawal policy on behalf of myself and my co-authors.